# PEDOT Films Doped with Titanyl Oxalate as Chemiresistive and Colorimetric Dual-Mode Sensors for the Detection of Hydrogen Peroxide Vapor

**DOI:** 10.3390/s23063120

**Published:** 2023-03-14

**Authors:** Xiaowen Xie, Nan Gao, Matthew Hunter, Ling Zhu, Xiaomei Yang, Shuai Chen, Ling Zang

**Affiliations:** 1Jiangxi Key Laboratory of Flexible Electronics and School of Pharmacy, Jiangxi Science & Technology Normal University, Nanchang 330013, China; 2Jiangxi Engineering Laboratory of Waterborne Coating, Nanchang 330013, China; 3Department of Materials Science and Engineering, University of Utah, Salt Lake City, UT 84112, USA; 4Nano Institute of Utah, University of Utah, Salt Lake City, UT 84112, USA

**Keywords:** hydrogen peroxide, chemosensor, chemiresistive, colorimetric, PEDOT:PSS, ammonium titanyl oxalate, humidity effect

## Abstract

Hydrogen peroxide (H_2_O_2_) is commonly used as an oxidizing, bleaching, or antiseptic agent. It is also hazardous at increased concentrations. It is therefore crucial to monitor the presence and concentration of H_2_O_2_, particularly in the vapor phase. However, it remains a challenge for many state-of-the-art chemical sensors (e.g., metal oxides) to detect hydrogen peroxide vapor (HPV) because of the interference of moisture in the form of humidity. Moisture, in the form of humidity, is guaranteed to be present in HPV to some extent. To meet this challenge, herein, we report a novel composite material based on poly(3,4-ethylenedioxythiophene):polystyrene sulfonate (PEDOT:PSS) doped with ammonium titanyl oxalate (ATO). This material can be fabricated as a thin film on electrode substrates for use in chemiresistive sensing of HPV. The adsorbed H_2_O_2_ will react with ATO, causing a colorimetric response in the material body. Combining colorimetric and chemiresistive responses resulted in a more reliable dual-function sensing method that improved the selectivity and sensitivity. Moreover, the composite film of PEDOT:PSS-ATO could be coated with a layer of pure PEDOT via in situ electrochemical synthesis. The pure PEDOT layer was hydrophobic, shielding the sensor material underneath from coming into contact with moisture. This was shown to mitigate the interference of humidity when detecting H_2_O_2_. A combination of these material properties makes the double-layer composite film, namely PEDOT:PSS-ATO/PEDOT, an ideal sensor platform for the detection of HPV. For example, upon a 9 min exposure to HPV at a concentration of 1.9 ppm, the electrical resistance of the film increased threefold, surpassing the bounds of the safety threshold. Meanwhile, the colorimetric response observed was 2.55 (defined as the color change ratio), a ratio at which the color change could be easily seen by the naked eye and quantified. We expect that this reported dual-mode sensor will find extensive practical applications in the fields of health and security with real-time, onsite monitoring of HPV.

## 1. Introduction

Oxidative gases, such as volatile peroxides (H_2_O_2_, peracetic acid, peroxide explosives, etc.) and ozone (O_3_), in the atmosphere are causes for concern in terms of biology, the general population’s health, safety, and environmental protection. H_2_O_2_ is the most common of these gases. It is a stable, reactive oxygen species in biological tissues [1] and exists as a colorless aqueous solution. It is an economical option for a variety of industries, and is found in oxidants, bleaching agents, and disinfectants [2,3]. Improper or excessive use of H_2_O_2_ can cause biological, health, or environmental harm [4,5]. Because H_2_O_2_ use is continually increasing, technology for the detection of HPV has become a research hotspot in the field of chemical sensors, and more broadly in material science and chemistry. There are many proven methods for the detection of H_2_O_2_, including titration, colorimetry, electrochemistry, enzymatic catalysis, chromatography, ultraviolet spectrophotometry, and analyses of fluorescence [6]. Each method has its own advantages and disadvantages. Electrochemical methods are the most common and require large, specialized instruments that are not ideal for on-site detection [7,8]. On the other hand, optical (colorimetric, fluorescent) sensors rely only on the relatively weak interactions between the analyte—H_2_O_2_ in this case—and the probing materials. They have the advantages of small-scale instrumentation, convenient on-site detection, the low price of production, and easy commercialization [9,10].

Due to the colorless and nearly odorless nature of HPV, it is difficult to distinguish HPV by smell and vision alone. Its guaranteed coexistence with moisture makes its detection extremely challenging. Humidity has been shown to greatly interfere with the performance of vapor-phase chemosensors [11]. H_2_O_2_ is miscible with water at all concentrations, guaranteeing an amount of moisture in HPV so long as the relative humidity is greater than zero; HPV does not exist as an independent phase. Aqueous solutions of H_2_O_2_ are used at concentrations of 1–3% in normal conditions for disinfecting and food processing purposes [12,13]. At concentrations of 30–35%, aqueous solutions of H_2_O_2_ are considered hazardous materials to people’s health [14,15]. These higher concentrations are found in industrial oxidants, bleaches, disinfectants, antiseptics, etc. Concentrations of 50% or higher are extremely uncommon, unless certain circumstances apply (military and aerospace applications). Thus, it is not practical to separate HPV from moisture or attempt mixing them at certain concentrations. Common gas mixture techniques such as pre-blending, absorbing the moisture with a desiccant, introducing additional moisture, and detection of the target gas are all not suitable for controlling the concentration of HPV.

In contrast to electric models, optical signals are less sensitive to humidity. A few fluorescent methods have been reported for the detection of HPV [16,17], but these methods present problems. The long detection reaction time, the time-consuming process of material selection and synthesis, difficulty in sampling, signal conversion, calibration of the complex instruments, and strong environmental interference are a few of these issues. Colorimetric strategies have been investigated as a different approach. They are simple, direct, have identifiable detection results that are visible to the naked eye, and have been shown to be a more promising direction than fluorescence methods [18]. Current colorimetric methods for sensing HPV exploit chemical reactions between H_2_O_2_ and the probing molecules in liquid phase systems [19]. However, as seen in gas-phase detection of active aldehydes [20], room temperature methods, such as electrochemical analysis and optical sensors, require far too much time for adsorption of the gas (hours) before any detection occurs. A colorimetric sensor based on ammonium titanyl oxalate (ATO), with HPV detection limits at a ppb level, has been reported [21]. However, the substrates and materials lacked long-term durability, making storage, optical testing, and real-time/online quantitative analysis difficult. In contrast, the sensors output electrical signals that can be easily modeled, quantitatively analyzed, and readily incorporated into device models. This is convenient for the development of portable and miniaturized chemosensors. However, these apparatuses are readily disturbed by environmental conditions, especially humidity, as well as other gaseous interference. Thus, fewer reports on the chemiresistive sensing of HPV have emerged. Nevertheless, an enzyme-based poly(3,4-ethylenedioxythiophene):polystyrene sulfonate (PEDOT:PSS)–horseradish peroxidase (HRP) paper-based chemiresistive sensor has been established [22]. The influence of variations in humidity on the HPV response, the sensor’s stability during the detection process, complex activities, long-term stability, and the loading process of HRP enzymes have still not been deeply discussed as of yet. In general, high humidity greatly affects the detection selectivity (cross-sensitivity), stability, reliability, and accuracy of all chemosensors, which is widely believed to be a core problem in their field of application. The oxidizing nature of HPV makes it relatively stable among the other gas phase components under ambient conditions. However, if chemically or biologically active substrates, such as HRP, are introduced into the gaseous sensing materials, their stability and their influence on detection must be considered. For example, chemiresistive sensors based on gas-sensitive materials (inorganic metal semiconductors) and carbon-based materials are susceptible to interference from other components in the air. This is because their response and sensitivity are highly dependent on factors such as the temperature, reactions at the modified surface interface, and micro- or nanostructure regulations, etc. [23,24,25]. Therefore, it is necessary to develop direct detection technologies targeting HPV under ambient conditions, relying on nonenzymatic organic probes and simple electric signal models.

Our group recently developed the first nonenzymatic chemiresistive HPV sensor that operates at room temperature [26]. The sensor relied on PEDOT:PSS, a widely known intrinsically conductive polymer (ICP) [27]. The possible applications of PEDOT:PSS-based films for detecting HPV functioned via basic electrical signal models. Such films also presented the possibility of detecting the chemically active gas phase analytes with varying levels of humidity. A PEDOT layer was also used to form a porous protective layer over the PEDOT:PSS surface, benefiting from their compatibility. With its abundant composite forms, PEDOT:PSS has been widely used in electrochemical and electronic sensors for detecting liquid-phase or gaseous chemical analytes [8,28]. In general, pristine PEDOT:PSS gives an inert response to most gaseous analytes in a normal atmosphere, even when using other models relying on patterned electrodes or arrays, etc. The most successful sensing method came from the combination of a complex model architecture and a composite material system design. Nevertheless, the real effect on or critical interference with a similar sensor’s response and the material’s lifetime will be from environmental humidity, the same serious issue encountered by its electronic applications. Of course, many gases are electronically detected by PEDOT:PSS-based sensors simply via moisture as a medium. Except for the coexisting moisture, the interference of the other common gas molecules in the scenario of an ambient atmosphere with HPV can be ignored. It was found that the basic chemiresistive model and the pristine PEDOT:PSS/PEDOT film were undesirable.

PEDOT-based films have excellent tunability of their inherent optical, electrical, and electrochemical properties by targeting oxidation as a form of chemical doping [8,29]. In this work, we committed to modulating probing film systems for improving their visualization of and fast response to HPV, relying on the special nature of HPV and the excellent optical transfer feature of PEDOT accompanied by the change in its electrical behavior under oxidation doping, and we displayed the special colorimetric selectivity of ATO for H_2_O_2_ with strong resistance to humidity [21]. The effects of ATO on the morphology, spectroscopy, stability under humidity, and the response of PEDOT:PSS-ATO and PEDOT:PSS-ATO/PEDOT films, as well as the resistance and colorimetric dual-signal response to HPV, were systematically studied. We believe such dual-model sensor systems can lead to more efficient chemiresistive sensors targeting HPV by withstanding the effects of the related humid atmosphere.

## 2. Experimental Section

### 2.1. Materials

Tetrabu-tylammonium hexafluorophosphate (Bu_4_NPF_6_, 98%) was purchased from Shanghai Energy Chemical Co., Ltd. (Shanghai, China). Ammonium titanyl oxalate (ATO, 99%) was purchased from Alfa Aesar (China) Chemical Co., Ltd. (Shanghai, China). Hydrogen peroxide (H_2_O_2_, aqueous solution, 30 wt.%) was purchased from Fuchen (Tianjin) Chemical Reagent Co., Ltd. (Tianjin, China). And 3,4-ethylenedioxythiophene (EDOT, 97%) and all organic solvents were purchased from J&K Scientific Ltd. (Shanghai, China). The aqueous dispersion of PEDOT:PSS was Clevios™ PH1000 (1.0–1.3 wt.%). Ultrapure water was made in the laboratory by an ultrapure water device (UPT-11-100T, Sichuan Ulupure Ultrapure Technology Co., Ltd., Sichuan, China). Indium-tin-oxide-coated glass (ITO-glass; item number KV-ITO-P001; sheet resistance, <10 Ω sq^−1^; transmittance, ≥83%; ITO sheet thickness, 174−182 nm) was purchased from Zhuhai Kaivo Optoelectronic Technology Co., Ltd. (Zhuhai, China).

### 2.2. Characterization

The EDOT was polymerized, and the electrochemical properties of the polymer were tested using an electrochemical workstation (Versa STAT 3, Princeton Applied Research, three-electrode system). Fourier transform infrared (FT-IR) spectra were recorded on a Bruker Vertex 70 FT-IR spectrometer (PerkinElmer Co., Ltd., Shanghai, China). The contact angle of each droplet was measured by a contact angle goniometer to determine the surface tension of the composite films (SDC-100, Dongguan Shengding Precision Instrument Co., Ltd., Dongguan, China) under ambient conditions of 25 ± 1 °C and 50% relative humidity (RH). The ultraviolet–visible (UV-vis) absorption spectra of all composite films were characterized by an UV-vis spectrophotometer (Specord Plus 200, Analytik Jena AG, Jena, Germany). The chemiresistive signal response of all the films to HPV was tested and recorded by a digital multimeter (DMM 6500 SourceMeter, Keithley, Beijing, China). The colorimetric signal response of all composite films to HPV was tested and recorded by a portable colorimeter (SC-10, Shenzhen Three-NH Technology Co., Ltd., Shenzhen, China).

### 2.3. Preparation of PEDOT-Based Composite Films

Similar to our previously published work [26], all PEDOT:PSS-based composite films were deposited on ITO substrates pre-cut into 10 × 20 × 1.1 mm substrates. Before use, the ITO substrates were treated by ultrasonic cleaning and then ultraviolet ozone cleaning. They were then washed with pure water; soaked in dichloromethane, acetone, ethanol, and pure water for 20 min; and then dried with high-purity N_2_. Finally, they were put into the ultraviolet ozone cleaning machine again with the ITO coating side upwards for 15 min, stored in pure isopropanol, and dried with pure N_2_ before use.

For preparation of the PEDOT:PSS-ATO films, 1 wt.% water-soluble ATO was directly added to the aqueous dispersion of PEDOT:PSS and mechanically stirred at 25 °C for 24 h. The cleaned ITO substrates were placed in the center of the suction cup of a desktop homogenizer, and 100 μL of a PEDOT:PSS-ATO water dispersion was placed on the glass surface until it was completely covered. Spin coating was performed at speeds of 1500 rpm for 30 s and 2500 rpm for 20 s. The steps above were repeated three times, and the coated films were then dried in a blast-drying oven at 50 °C for 2 h. The PEDOT:PSS-ATO films were ready for characterization at this point (Figure 1).

For preparation of the PEDOT:PSS-ATO/PEDOT film, a three-electrode electrochemical system was built for the electrochemical polymerization of EDOT (1 mM in acetonitrile/tetrabutylammonium hexafluorophosphate (0.1 M) electrolyte) by chronocoulometry at 25 °C, in which a PEDOT:PSS-ATO film electrode, prepared as described above, was used as a working electrode, a platinum (Pt) sheet was used as the counter electrode, and a Ag/AgCl wire was used as the reference electrode. The optimal electrosynthesized amount (i.e., its deposition thickness) of the PEDOT layer was controlled to be 30 mC. The PEDOT:PSS-ATO/PEDOT films were ready for characterization at this point (Figure 1).

For comparison, PEDOT:PSS, PEDOT, and PEDOT:PSS/PEDOT were prepared following the same drop-coating and electrochemical polymerization deposition methods as mentioned above. ATO was omitted from these syntheses.

### 2.4. Construction of the HPV Detection System

The setup used for the generation and detection of HPV, as shown in Figure 1, was the same as the system built in our previously published work [26]. In view of the safety threshold value of 1 ppm and the short-term contact limit of 2 ppm within a 15 min timeframe [30], four systems were created containing 10.5 ppm, 4.0 ppm, 1.9 ppm, and 1.0 ppm of HPV. The concentrations were determined using saturated vapor pressures. As discussed above, unlike other gaseous analytes, simultaneously maintaining fixed humidity levels and HPV concentrations is difficult. Limiting the interference of humidity, instead of a fixed concentration level, seems more practical.

## 3. Results and Discussion

### 3.1. Chemiresistive HPV Sensing Based on PEDOT:PSS-ATO and PEDOT:PSS-ATO/PEDOT Films

To verify the differences in the responses to humidity and HPV, several mutually verifiable anti-interference experiments were carried out. All experiments consisted of at least six parallel runs for verification. Analysis of the response time and the response/recovery behavior aided in identifying distinct responses within minutes. This was represented by the time required to reach maximum resistance or a change in color in an equilibrium state. Recovery was practically infeasible due to the swelling effect of moisture and the chemical interaction of H_2_O_2_ with the sensor film. As reported in our previous work [26], the adsorption of moisture causes the PSS chains to swell, and this changes the electrical resistance of PEDOT:PSS films. Since the evaporation of water takes time, this resulted in significant extension of the signal recovery time depending on the humidity level in the environment. The chemiresistive sensing response is primarily due to the electronic interaction between H_2_O_2_ and the positive dopant (hole) of PEDOT. Such an interaction may transform to a permanent reaction by converting H_2_O_2_ to oxygen, which, in turn, makes holes in PEDOT (i.e., causes a decrease in conductivity). To recover the reduced conductivity, appropriate reagents can be added as secondary dopants to regenerate holes [31]. However, these methods are practically infeasible regarding their application in sensors, and may create new contamination and dangerous operations. It should be mentioned that for efficient sensing, the sensor–analyte signal response is of equal importance for the subsequent adsorption of the analyte’s molecules to the probing film. This is why PEDOT:PSS in ICPs was selected in this work. The balance between the humidity-assisted adsorption and its interference with the signal response to HPV must be carefully regulated. Our goal was to develop a portable sensor for detecting indoor or residual HPV after the use of hydrogen peroxide in its wide variety of industrial uses. The system proved suitable for simulating real-world cases of hydrogen peroxide exposure.

The effect of ATO on the structure of PEDOT:PSS-ATO films was tested by FT-IR spectroscopy. As shown in Figure 2a, compared with PEDOT:PSS, a new broad band at 3130 cm^−1^ could be found. This indicated the stretching vibration of N-H, while the spectrum at 1720 cm^−1^ and 1250 cm^−1^ belonged to different vibration modes from the oxalic acid group. The band at 765 cm^−1^ represented the tensile vibration of Ti-O [32], which proved that ATO had been effectively compounded into the PEDOT:PSS film. This may be ascribed to its good water solubility and the hydrogen-bonded molecular structure of ammonium oxalate and titanyl. Meanwhile, as shown in Figure 2b, the UV-vis absorbance of the PEDOT:PSS film was significantly higher than that of the PEDOT:PSS-ATO films, especially at 500–600 nm. This also corresponded to their dark blue color, as the ATO combination weakens the blue color of PEDOT:PSS. Its own discoloration was identifiable following an oxidative reaction with H_2_O_2_. This is a relevant factor to consider, as PEDOT itself is an oxidation (doping)-driven conductor (Appendix A).

Similar to our previously reported resistance–time curves of PEDOT:PSS and PEDOT:PSS/PEDOT films [26], under a continuous increase in the chamber’s humidity, the PEDOT:PSS-ATO film exhibited a typical humidity response, as shown in Figure 2c. As the humidity increased with time, its resistance increased quickly until it reached a maximum of about 50–60% RH (7.6 times the initial value). This happened within 20 min. The film’s resistance continued to decrease until it reached the same level as the initial value. This took around 40 min. In comparison, the PEDOT:PSS-ATO/PEDOT film showed much more stable resistance within the moisture-rich environment. The fluctuations were less than 10 Ω within 40 min (Figure 2d), which was consistent with the behaviors of the PEDOT:PSS/PEDOT film. These results again proved that a protective layer of PEDOT can effectively reduce the influence of moisture on PSS swelling, thus improving the moisture resistance of the composite films. This was further demonstrated by the surface wettability test shown in Figure 3. The contact angle of the PEDOT:PSS (33°) was hardly changed by the combination of ATO. Obvious differences emerged after electrochemical deposition of a PEDOT layer on their surfaces. The contact angles of PEDOT:PSS/PEDOT films gradually increased with an increase in the quantity of the imposed electric charge from 10 mC to 30 mC during the electropolymerization of EDOT. Promisingly, PEDOT:PSS/PEDOT (30 mC) and PEDOT:PSS-ATO/PEDOT films (30 mC) showed obvious hydrophobic characteristics (water angles of >90 °C). For this reason, all PEDOT-coated composite films used for sensing research in this study were electrosynthesized using a charge of 30 mC, unless otherwise stated. Contact angle measurements are advantageous, as they laterally evaluate the influence of the film’s wettability on small droplets formed by the moisture on the sensor film during the detection of HPV.

The resistance response results of PEDOT:PSS-ATO and PEDOT:PSS-ATO/PEDOT films exposed to different concentrations of HPV are shown in Appendix A. It was found that the overall trend of the resistance–time response (Appendix A) of the PEDOT:PSS-ATO film was similar to that of the PEDOT:PSS film [26]. Once exposed to HPV, the film’s resistance changed rapidly within seconds and rose quickly from the initial values (~97 Ω, slightly higher than that of the pristine PEDOT:PSS film (~87 Ω)) to 10^2^ Ω [26]. The weak, but limited, effect of ATO on the PEDOT:PSS film’s conductivity can be ascribed to the presence of ammonium oxalate groups in its molecular structure. They interact with PSS^−^ chains to change the degree of doping of PEDOT^+^ or drive the recombination of the molecular chains. Within about 5–20 min, the values were relatively high and stable. They then decreased after 5 min to a similar level to their initial values. In view of the effect of the concentrations of HPV, the resistance responses (ΔR/R_0_) of PEDOT:PSS-ATO films were 8.52 (10.5 ppm), 5.93 (4.0 ppm), 3.06 (1.9 ppm), and 2.28 (1.0 ppm). They reached their highest peaks at 16 min, 18 min, 18 min, and 17 min, respectively. Greater concentrations of the analyte resulted in a stronger signal contrast without a significant difference in the peak time.

To analyze the behavior of PEDOT:PSS-ATO films during the detection of HPV in detail, a three-stage explanation (as shown in Appendix A) is necessary. First, the strong moisture adsorption due to the presence of hygroscopic PSS^−^ led to a decrease om the electrical interconnection between the PEDOT^+^ chains [33,34] due to the P-type semiconductor’s nature (rich in positive holes) of PEDOT-based ICPs [27]. While H_2_O_2_ is commonly used as an oxidant (with itself being reduced into water), it can also act as a reducing agent or an electron donor (with itself being oxidized into oxygen), depending on the reagent it reacts with. When exposed to H_2_O_2_, an electron transfer interaction occurs between the H_2_O_2_ and PEDOT, with H_2_O_2_ occupying holes in the PEDOT chain, which, in turn, reduces the doping density, resulting in a decrease in the electrical conductivity (or an increase in resistance). In addition, the organic structure of the PEDOT molecule itself suffered from the oxidation of H_2_O_2_ [35]. In the second stage, with the saturation of the adsorbed moisture, the swelling of the polymer chain was suppressed. The continual increase in the chamber’s humidity gradually condensed the moisture into small water droplets on the surface of the sensor film. This environment could dissolve PSS^−^, resulting in more ionic conductivity. This formed a competitive relationship, with the positive effect of H_2_O_2_ on resistance. Thirdly, as the humidity gradually increased with the extension of the detection time, further contact with H_2_O_2_ in the atmosphere was prevented due to a water meniscus forming on the film’s surface. This caused a decrease in the resistance. In contrast to the previously reported PEDOT:PSS film [26], the PEDOT:PSS-ATO film had a much shorter peak time but a lower resistance response (∆R/R_0_) in the range of 1.9 ppm to 10.5 ppm HPV (Table 1). Taking 4.0 ppm as an example, the PEDOT:PSS film sensor reached its first resistance response peak in about 24 min, while the PEDOT:PSS-ATO film did so in about 18 min. The latter’s response at the peak time was 7% less in magnitude. This may be related to the reaction between ATO and H_2_O_2_, which consumes HPV and generates H_2_O simultaneously. This accelerates the detection process. The ExpAssoc function (y= 1.77636 × 10^−5^ + 9.12598 × (1 − e^(−x/3.87165)^), R^2^ = 0.99384) as used fitting the data of the resistance response and the HPV concentration, and the fitted curve can be seen in Appendix A.

The trend shown in the resistance–time response curves of PEDOT:PSS-ATO/PEDOT films in response to HPV at different concentrations (Figure 4a) is similar to that of PEDOT:PSS/PEDOT films [26]. After exposure to HPV, the film’s resistance values rose rapidly to a series of peak values. They started at an initial level of about 53 Ω, which was similar to 48 Ω seen for the PEDOT:PSS/PEDOT film. This was 1.83 times lower than that of the PEDOT:PSS-ATO film, caused by the higher conductivity of PEDOT:PSS and showcasing the more efficient PSS^−^ to PEDOT^+^ doping system. With time, the resistance values decreased until reaching similar levels to their initial values after 20 min. The resistance responses (∆R/R_0_) were calculated by the peak values, which were 5.68 (10.5 ppm), 4.42 (4.0 ppm), 2.19 (1.9 ppm), and 0.30 (1.0 ppm). The ExpAssoc function (y = 5.75634 × (1 − e^(−x/2.42891)^), R^2^ = 0.85399) was used to fit the resistance response and HPV concentration data. The resulting fitted curve can be seen in Figure 4b. By comparing the resistance response data of PEDOT:PSS-ATO/PEDOT films with those of PEDOT:PSS-ATO [26], it was seen that the latter’s response was smaller than that of the former. The films containing ATO reached their respective peak values (about 14 min for 1.0 ppm, 13 min for 1.9 ppm, 9 min for 4.0 ppm, and 10 min for 10.5 ppm) much more quickly. Here, it should be stated that for 10.5 ppm HPV, the resistance quickly reached a maximum level within 5 min, changing only incrementally around its peak value, until around 18 min. The time-to-peak of the PEDOT:PSS-ATO/PEDOT film was reduced by about 50% for 1.9 ppm. The reason for these behaviors may be that, beyond the selective reaction between ATO and H_2_O_2_, the generated H_2_O could be locked in the inner layer of PEDOT:PSS-ATO when covered by the PEDOT layer. This accelerated the detection process. The PEDOT:PSS-ATO layer may have been affected by small water droplets forming, which were caused by condensation, undergoing a transition from the Cassie state to the Wenzel state, on the surface of the outer layer of PEDOT [36,37]. If we take 1.0 ppm as an example, the resistance value of the PEDOT:PSS/PEDOT film continued to rise for 50 min [26], while the resistance value of the PEDOT:PSS-ATO/PEDOT film peaked at about 10 min and then decreased slowly (Figure 4a (inset)).

### 3.2. Colorimetric HPV Sensing Based on PEDOT:PSS-ATO and PEDOT:PSS-ATO/PEDOT Films

From the data presented in Table 1, we can conclude that the combination of the non-conductive ATO component weakened the change in the resistance of PEDOT:PSS when exposed to HPV. The response times, however, became faster. The generation and dispersion of HPV will inevitably form small liquid droplets on the film’s surface. Such atomization tendencies are a necessary property of HPV for certain applications, such as disinfection. As a moisture-associated gas analyte, the effects of high humidity and the tendency of condensed droplets to form on the sensor film on the detection of HPV have to be seriously considered. Promisingly, beyond the discussed electrical model, colorimetric functions are less prone to disturbance by humidity. Colorimetric detection methods are an important direction of development for popularization of the sensors because results that are visible to the naked eye can be obtained in real time [18]. For the detection of HPV, optical detection methods are based on the chemical reaction between H_2_O_2_ and the sensing molecules in liquid phase systems [19,38]. The literature reports [21] a simple, low-cost HPV sensor developed on the basis of highly selective complexation-induced discoloration effect between colorless ATO and H_2_O_2_. Benefiting from this rapid chemochromic reaction, with the excellent optical transfer (color and transparency) behavior of PEDOT during interaction with H_2_O_2_, we further investigated the colorimetric signal response of PEDOT-based films to HPV while simultaneously monitoring changes in the resistance.

All the sensor films were tested using a colorimeter before implanting them into the HPV sensing chamber to establish a control. After the detection tests were completed, the films were carefully taken out of the chamber and left for 10 min before the films’ colors were recorded. To keep the colors undistorted, the films to be tested were placed on white A4 paper for collection of the colorimetric data. The beam of the portable colorimeter was illuminated from the back of the ITO substrates, and the results of the color difference test were outputted as ΔL, Δa, and Δb values and evaluated by the ΔE values.

ΔE is the comprehensive difference in total color, which can be calculated by the following formula [39]:ΔE = [(ΔL)^2^ + (Δa)^2^ + (Δb)^2^]^1/2^(1)

ΔL > 0 indicates that the test sample was whiter than the standard sample. Otherwise, it was assumed to be darker. Because the sample was placed on white A4 paper for testing, ΔL > 0 means that the test sample was lighter and more transparent than the standard sample. Δa > 0 indicates that the test sample was redder than the standard sample, and vice versa. Δb > 0 indicates that the test sample was yellower than the standard sample, and vice versa. The greater the ΔE value, the greater the color difference of the sensor film after the detection of HPV. When ΔE is between 0 and 1, the color difference was minimal and was not distinguishable by the naked eye. When ΔE is between 1 and 2, the color difference was faintly recognized by the naked eye. When ΔE is between 2 and 3, the colorimetric response of the film was more obvious, and the color difference was distinguishable by the naked eye. When ΔE is between 3 and 5, the color difference was very distinguishable. When ΔE > 5, the results resembled two separate colors.

With exposure to increasing concentrations of HPV, PEDOT:PSS-ATO and PEDOT:PSS-ATO/PEDOT films showed obvious color changes from blue to yellow-green, as seen in Figure 5a,b. The data shown in Figure 5 and listed in Table 2 show the greater response (ΔE) of the colorimetric response–HPV concentration as a fitted curve. In light of the blue color in these films caused by PEDOT, the pristine PEDOT:PSS-ATO/PEDOT film showed much darker color than the PEDOT:PSS-ATO film. However, the additional polymer layer could weaken the colorimetric response due to the issues of thickness and background color. Nevertheless, by adding colorless ATO, the consequent oxidative discoloration to yellow (Figure 5a (inset)) made these films exhibit a more distinct and abundant color change than PEDOT:PSS (Appendix A) and PEDOT:PSS/PEDOT (Appendix A) films. These only monochromatically changed colors of blue, due to the well-known doping-driven electrical and optical mechanisms of PEDOT (Appendix A).

With HPV concentrations as low as 1.0 ppm, the colorimetric response of PEDOT:PSS-ATO and PEDOT:PSS-ATO/PEDOT films achieved a color difference of up to 2.6 and 1.57, respectively; this was visible to the naked eye. As the HPV concentration increased, the ΔL value of the PEDOT:PSS-ATO film increased, indicating that HPV caused the film’s color change to be more obvious with a decreasing Δa value and an increasing Δb value. That is, the color of the film changed to be more green and yellow, not staying a primary blue. Compared with the PEDOT:PSS (Appendix A) film, after being exposed to HPV, the ΔL value of the PEDOT:PSS-ATO/PEDOT film increased, the Δa value decreased, and the overall trend of Δb increased. This was seen as the film’s color changed to shades of green and yellow, not to shades of blue. As shown in Appendix A and Table 2, although the value of the comprehensive color difference (ΔE) of the PEDOT and PEDOT:PSS/PEDOT films increased as the HPV concentrations rose from 1.0 ppm to 10.5 ppm, all their ΔE values were less than 1. This means that the color change was almost invisible to the naked eye. Thus, it should be said that coating an outer layer of PEDOT on the surface of PEDOT:PSS film had a negative effect on its colorimetric response. The reason may be that the PEDOT layer hindered the entry of H_2_O_2_ into the PEDOT:PSS or PEDOT:PSS-ATO, not just shielding them from moisture. The colorimetric response–HPV concentration curves were fitted by the ExpAssoc function for PEDOT:PSS-ATO (y = 8.88178 × 10^−16^ + 7.5728 × (1 − e^(−x/1.92937)^), R^2^ = 0.96464) and PEDOT:PSS-ATO/PEDOT (y = 4.44089 × 10^−16^ + 3.3662 × (1 − e^(−x/2.1625)^), R^2^ = 0.9367) films. This trend was similar to the electrical curve shown in Figure 4b, demonstrating the synergy and consistency of a dual-model response while also proving the rationality of the three-stage response processes described above. The cooperative relationship between the color and resistance signals enhanced the accuracy and reliability of these sensing systems.

To further investigate the colorimetric response, the UV-vis absorption spectra of PEDOT:PSS-ATO and PEDOT:PSS-ATO/PEDOT films were tested before and after the detection of HPV. The results are shown in Figure 5c,d. After sensing HPV, PEDOT:PSS-ATO films showed an increase in absorbance around 400 nm. This value increased as the HPV concentration increased. Such absorbance changes came from the highly selective reaction between the Ti(IV)oxycarbonyl unit in ATO with H_2_O_2_, forming Ti(IV)-peroxide bonds. This reaction made the complex film change from colorless to bright yellow (Figure 5a (inset)) [21]. Similarly, in the UV-vis absorption spectra of PEDOT:PSS-ATO/PEDOT films after the detection of HPV, it was also seen that the absorbance at around 400 nm increased with the concentration of HPV. The decrease in absorbance from around 500 nm to 600 nm indicated that the color of the film became less blue as the HPV concentration increased. This remains congruent with the results of the color change phenomenon and the Δb analysis. Both the PEDOT:PSS [26] and PEDOT:PSS/PEDOT (Appendix A) films showed a similar absorbance phenomenon at 900 nm. That is, the absorbance decreased with increasing HPV concentrations. This is consistent with the analysis of the resistance response and the higher ΔL values. This is because the spectral absorption at about 900 nm represents the concentration of bipolarons on the PEDOT^+^ chain, which directly affected the film’s conductivity. The higher the concentration of bipolarons, the higher the values of conductivity will be. As shown here, the decreasing trend of the concentration of bipolarons in the composite films during the HPV detection process corresponded to the trend of the increasing resistance response values [20].

## 4. Conclusions and Prospects

In this paper, PEDOT:PSS-ATO and PEDOT:PSS-ATO/PEDOT films were prepared by adding colorless ATO to PEDOT:PSS via aqueous dispersions through mechanical blending methods. ATO has a known characteristic complexation-induced discoloration upon exposure to H_2_O_2_. The films were used for chemiresistive and colorimetric sensing of HPV in the presence of high humidity. The films showed highly sensitive and selective monitoring of the electrical resistance, colorimetric response behavior, and structural stability under different concentrations of HPV. The controllable interfering effect of humidity during detection was analyzed by an on-site digital multimeter, a portable colorimeter, FT-IR, UV-vis, contact angle measurements, and other analysis methods. In comparison with PEDOT:PSS and PEDOT:PSS/PEDOT films, PEDOT:PSS-ATO/PEDOT films are believed to support a much more promising dual-mode response to HPV. The device composed of PEDOT:PSS and ATO, with a porous and hydrophobic protective layer of electrochemically deposited PEDOT, exploited the selective chemical reaction when ATO was exposed to H_2_O_2_. They also capitalized on the comprehensive structural, electrical, and optical advantages of PEDOT-based films. This study not only solved the long-term issue of HPV detection but has also provided material and theoretical support for the future development of a practical HPV sensor device. This work also gives insight for the exploration of humidity-resistive chemosensors based on organic film probes for ambient conditions. It can also promote the promising application of PEDOT:PSS-based composite materials in electronic and even electrochemical gas-sensing applications.

There are still many challenges to overcome for the technologies presented in this work. For example, the introduction of ATO impaired the films’ chemiresistive response, and the PEDOT outer layer masked the ATO’s color response. Future work could concern itself with optimizing the generation and dispersion of HPV, a method for measuring the signal, electronic models, device architecture, a composite material system to inhibit the polymer from swelling under humid conditions, and the films’ optic/electronic performances. PEDOT:PSS is highly tunable [8,40,41]. Optimization studies could begin by employing secondary dopants such as the volatile organic reagents dimethyl sulfoxide (DMSO) and ethylene glycol (EG) to enhance the sensitivity of the response by substantially increasing the films’ initial conductivity, or constructing covalent crosslinking systems with polyvinyl alcohol (PVA) and glutaraldehyde (GA) to improve the films’ stability and immunity to interference under varying levels of humidity. These modification attempts will expand the scope of knowledge necessary to expand this work. The development of a portable sensor system (replaceable sensing film strips) for detecting residual indoor HPV after the use of hydrogen peroxide could lead to broader applications of PEDOT:PSS chemiresistive sensors from the detection of environmental HPV to monitoring trace amounts of H_2_O_2_ in explosive volatiles in the security field.

## Figures and Tables

**Figure 1 sensors-23-03120-f001:**
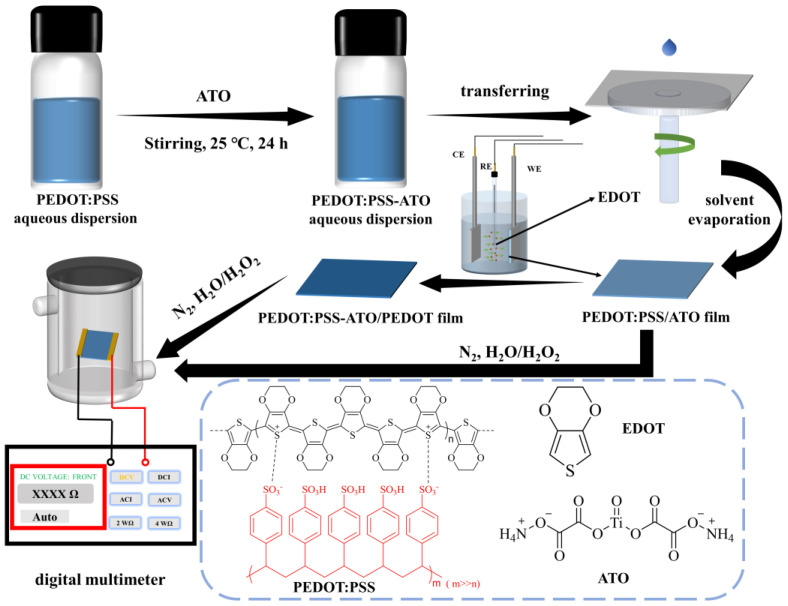
Schematic of the preparation and detection process of PEDOT:PSS-ATO and PEDOT:PSS-ATO/PEDOT films. CE: counter electrode; RE: reference electrode; WE: working electrode.

**Figure 2 sensors-23-03120-f002:**
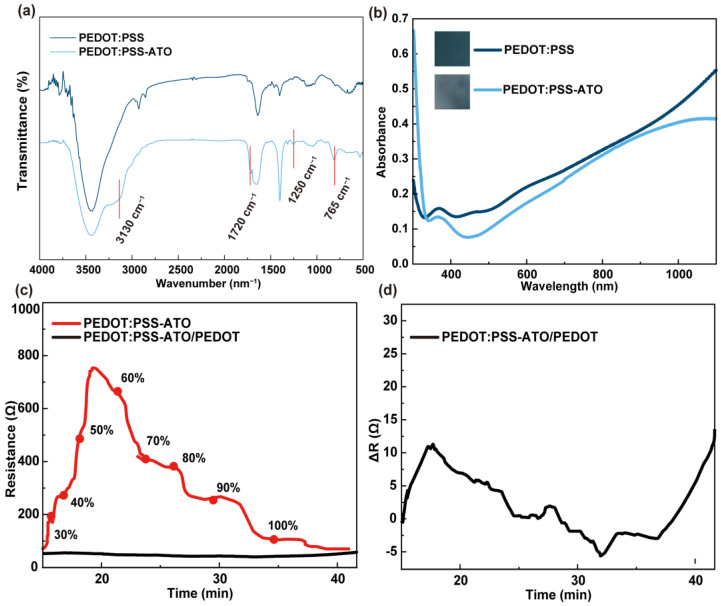
(**a**) Fourier transform spectra of PEDOT:PSS and PEDOT:PSS-ATO films. (**b**) UV-visible absorption spectra of PEDOT:PSS and PEDOT:PSS-ATO films (inset: photos of the film). Resistance–time curves of (**c**) PEDOT:PSS-ATO and PEDOT:PSS-ATO/PEDOT films under increasing levels of chamber humidity (30–100% RH). (**d**) ΔR of PEDOT:PSS-ATO/PEDOT film under increasing levels of chamber humidity (30–100% RH).

**Figure 3 sensors-23-03120-f003:**
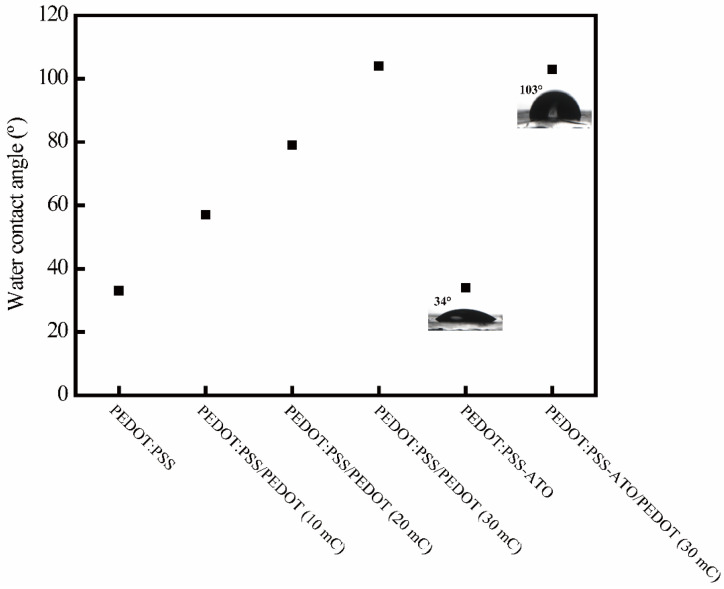
Water contact angles of PEDOT-based films and (inset) images of a droplet on the surface of PEDOT:PSS-ATO and PEDOT:PSS-ATO/PEDOT films.

**Figure 4 sensors-23-03120-f004:**
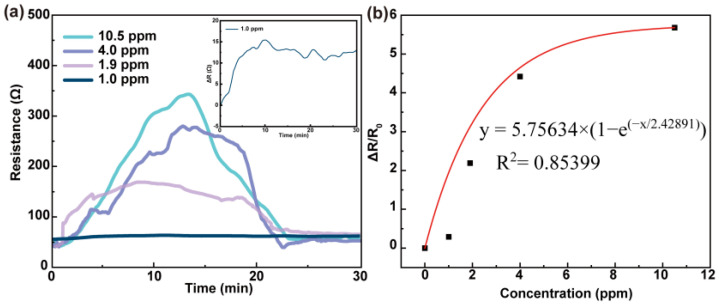
(**a**) Resistance–time curves of PEDOT:PSS-ATO/PEDOT films in response to different concentrations of HPV (inset: an enlarged version of the curve for 1.0 ppm). (**b**) Resistance response–HPV concentration fitting curve of PEDOT:PSS-ATO/PEDOT films.

**Figure 5 sensors-23-03120-f005:**
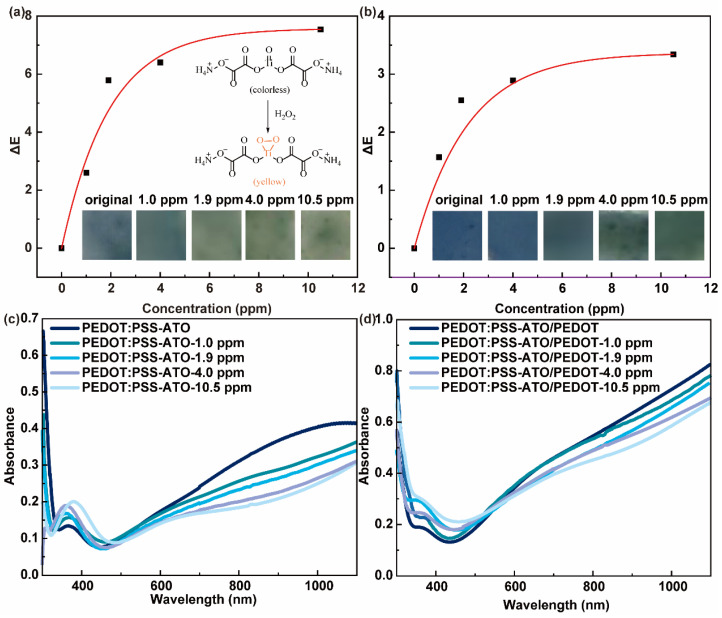
Colorimetric response–HPV concentration fitting curves of (**a**) PEDOT:PSS-ATO and (**b**) PEDOT:PSS-ATO/PEDOT films (inset: mechanism of the discoloration of ATO to H_2_O_2_ through oxidation, showing photos of the film before and after exposure to different concentrations of HPV). UV-visible absorption spectra of the original (**c**) PEDOT:PSS-ATO and (**d**) PEDOT:PSS-ATO/PEDOT films and after the detection of different concentrations of HPV.

**Table 1 sensors-23-03120-t001:** Resistance response data of PEDOT:PSS, PEDOT:PSS/PEDOT, PEDOT:PSS-ATO, and PEDOT:PSS-ATO/PEDOT films under different concentrations of HPV.

Sensor Films	HPV (ppm) Response (ΔR/R_0_)				
	1.0	1.9	4.0	10.5	
PEDOT:PSS	0.54	5.64	8.51	10.87	[26]
PEDOT:PSS/PEDOT	0.72	5.98	13.82	17.73	
PEDOT:PSS-ATO	2.28	3.06	5.93	8.52	This work
PEDOT:PSS-ATO/PEDOT	0.29	2.19	4.42	3.68	This work

**Table 2 sensors-23-03120-t002:** Colorimetric response results of PEDOT:PSS, PEDOT, PEDOT:PSS/PEDOT, PEDOT:PSS-ATO, and PEDOT:PSS-ATO/PEDOT films under different concentrations of HPV.

Films	HPV (ppm)	ΔL	Δa	Δb	ΔE
PEDOT:PSS	1.0	2.05	−0.46	−0.18	2.11
	1.9	3.02	−0.15	0.18	3.03
	4.0	3.99	−1.17	0.77	4.23
	10.5	4.58	−1.20	2.00	5.13
PEDOT	1.0	0.04	0.08	−0.12	0.15
	1.9	0	−0.03	0.23	0.23
	4.0	−0.51	0.11	0.45	0.69
	10.5	0.74	−0.03	0.24	0.78
PEDOT:PSSS/PEDOT	1.0	0.51	0.11	−0.45	0.69
	1.9	0.88	0.91	−0.83	1.51
	4.0	1.68	0.72	−0.04	1.82
	10.5	1.87	0.58	0.7	2.08
PEDOT:PSS-ATO	1.0	0.73	0.15	2.49	2.60
	1.9	1.99	−1.04	5.34	5.79
	4.0	2.15	−0.29	6.02	6.40
	10.5	2.83	−1.36	6.85	7.54
PEDOT:PSS-ATO/PEDOT	1.0	−1.35	0.36	−0.7	1.57
	1.9	1.76	−0.81	1.65	2.55
	4.0	2.22	−1.05	1.52	2.89
	10.5	2.15	−0.39	2.52	3.34

## Data Availability

Not applicable.

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
