# Peer review of "PEDOT Films Doped with Titanyl Oxalate as Chemiresistive and Colorimetric Dual-Mode Sensors for the Detection of Hydrogen Peroxide Vapor"

_sensors, 2023, doi:10.3390/s23063120_

Round 1

Reviewer 1 Report

Reviewer: In the present work, authors have reported the dual mode sensor based PEDOT:PSS-ATO and PEDOT:PSS-ATO/PEDOT composite, and further investigated the structure as well as chemiresistive and colorimetric sensing characteristics for H2O2 vapors. The results revealed that PEDOT is used as hydrophobic captive layer which have certainly affected the response characteristics of sensors. In my opinion, there are some major issues in the manuscript which needs to be addressed before it would be accepted in “sensors” journal.

1.      “Aqueous solution of H2O2 is used at concentrations of 1 ~ 3% in normal conditions for disinfection and food processing. At 30 ~ 35% concentrations, aqueous solutions of H2O2 are considered hazardous materials.” Please add the reference for this statement.

2.      Some of the references in the manuscript do not match with statement such as reference [22]. This effects the argument of author. Please check mistakes like these and correct it. 

3.      Figure 2c showed the humidity detection response of the sensors, please explain the reason behind the decreased resistance after reaching the maximum value.

4.      The study of bandgap should be included in correspondence with absorbance results, and further effect of change in bandgap on gas sensing should be included.

5.      It is mentioned in the manuscript that HPV has oxidizing nature and later in the manuscript, while explaining the sensing mechanism it is reported as electron donor. These both statement contradicts with each other. Please explain these statements.

6.      It is mentioned in manuscript that, due to coating of PEDOT hydrophobic layer on sensor, PEDOT:PSS-ATO layer may have been affected by the small water droplets. But, it seems like PEDOT:PSS-ATO/PEDOT showed fast recovery as compared to bare sensor where bigger water droplet acts as protective layer and decrease the interaction of H2O2 with sensor and resistance of the sensor decreases. Please explain the reasons more thoroughly.

7.      The explanation for why recovery is difficult due to the chemical reaction of H2O2 is insufficient. Adding few more reasons would provide a more comprehensive understanding of the issue.        

8.      In colorimetric sensing, if PDOT layer acts as shielding layer than how can the concentration of H2O2can be controlled.

9.      It would be helpful to also add the transmittance information in addition to the absorbance information.

10.  In the cases of Figure 5 (C), (D) The lines in the graph are not color-coded well. Please use the distinct colors for these figures.

11.  The information about the thickness of ITO (Indium Tin Oxide) in ITO substrates is not provided. Because it is important factor of chemiresistive.

12.  It would be helpful to also graph the transmittance information in addition to the absorbance information.

13.  It would be nice to have data on HPV sensors that work at other temperatures than just room temperature as well.

14.  The explanation for why recovery is difficult due to the chemical reaction of H2O2 is insufficient. Adding two or more reasons would provide a more comprehensive understanding of the issue.

Reviewer 2 Report

Xie xiaowen et.al. investigated the sensing performance of PEDOT:PSS-ATO/PEDOT sample toward H2O2 from chemiresistive and colorimetric modes. The effect of different componential combinations was unveiled and some characterization techniques were adopted for performance analysis. The work seems interesting but there are still some issues to be well addressed before the reconsideration of this work.

1) The authors should specify the detailed application scenarios for their investigation, particular for the adopted H2O2 concentration. Also some recent work about PEDOT:PSS based chemiresistive sensing should be added such as Nanotechnology, 2022, 33, 065501.

2) Label the vertical axis in Fig. 2a.

3) For chemiresistive sensing, the resistance barely attained a stable state upon H2O2 adsorption. Give the possible reasons behind this.

4) The authors should provide previous work of other groups to showcase the advances of their efforts.

5) The structure-property relationship is very close. In this work, morphological or structural features of the as-prepared samples were obscure and not well linked.

Round 2

Reviewer 1 Report

All of my concerns are resolved, so I accept the article to publish in this journal.  

Reviewer 2 Report

The authors have provided a careful response to all issues. Thus, the current manuscript could be accepted.